# Alleviation of neuronal energy deficiency by mTOR inhibition as a treatment for mitochondria-related neurodegeneration

Xinde Zheng[1†], Leah Boyer[2†], Mingji Jin[1], Yongsung Kim[2], Weiwei Fan[3], Cedric Bardy[2], Travis Berggren[4], Ronald M Evans[3,5], Fred H Gage[2*], Tony Hunter[1*]

[1]Molecular and Cell Biology Laboratory, Salk Institute for Biological Studies, La Jolla, United States; [2]Laboratory of Genetics, Salk Institute for Biological Studies, La Jolla, United States; [3]Gene Expression Laboratory, Salk Institute for Biological Studies, La Jolla, United States; [4]Stem Cell Core, Salk Institute for Biological Studies, La Jolla, United States; [5]Howard Hughes Medical Institute, Salk Institute for Biological Studies, La Jolla, United States

**Abstract** mTOR inhibition is beneficial in neurodegenerative disease models and its effects are often attributable to the modulation of autophagy and anti-apoptosis. Here, we report a neglected but important bioenergetic effect of mTOR inhibition in neurons. mTOR inhibition by rapamycin significantly preserves neuronal ATP levels, particularly when oxidative phosphorylation is impaired, such as in neurons treated with mitochondrial inhibitors, or in neurons derived from maternally inherited Leigh syndrome (MILS) patient iPS cells with ATP synthase deficiency. Rapamycin treatment significantly improves the resistance of MILS neurons to glutamate toxicity. Surprisingly, in mitochondrially defective neurons, but not neuroprogenitor cells, ribosomal S6 and S6 kinase phosphorylation increased over time, despite activation of AMPK, which is often linked to mTOR inhibition. A rapamycin-induced decrease in protein synthesis, a major energy-consuming process, may account for its ATP-saving effect. We propose that a mild reduction in protein synthesis may have the potential to treat mitochondria-related neurodegeneration.

*For correspondence: gage@salk.edu (FHG); hunter@salk.edu (TH)

[†]These authors contributed equally to this work

## Introduction

The mTOR complexes coordinate nutrient availability with cell growth and proliferation, promoting protein synthesis and inhibiting autophagy (*Laplante and Sabatini, 2012*). Protein homeostasis is often distorted in neurodegenerative diseases, such as Parkinson's and Alzheimer's disease, as well as PolyQ and other proteinopathies, making mTOR an attractive therapeutic target (*Bové et al., 2011*). Studies from animal models support mTOR inhibition as a promising therapeutic approach for neurodegenerative diseases, although several distinct beneficial mechanisms have been proposed. Rapamycin, an mTORC1 inhibitor, reduces neuronal cell death in a mouse model of Parkinson's disease, and decreased synthesis of DDIT4 (DNA-damage-inducible transcript 4) was proposed to provide the protective effect by maintaining AKT pro-survival phosphorylation (*Malagelada et al., 2010*). Rapamycin strongly suppresses degeneration of dopaminergic neurons in *Drosophila* with loss of function mutations of *PINK1* and *PARKIN*, genes in which mutations cause human early onset Parkinsonism; importantly, overexpression of 4E-BP, a protein synthesis inhibitor downstream of the mTORC1 pathway, also rescues neuronal degeneration in these fly mutants. Increased production of GSTS1, a detoxifying enzyme, was suggested to be the beneficial factor (*Tain et al, 2009*). In a

**eLife digest** Living cells need to maintain an optimal balance between making new proteins and destroying older ones. Building proteins requires a supply of nutrients and appropriate levels of energy, and mammalian cells rely on a protein called mTOR to sense both nutrient availability and energy levels. Nutrients activate mTOR signaling to promote protein synthesis. In contrast, a lack of nutrients and low energy levels inhibit mTOR, which slows down protein synthesis to help the cell to conserve vital resources.

The balance between protein synthesis and degradation is often perturbed in diseases that involve the progressive loss of nerve cells, and a drug called rapamycin – which inhibits mTOR signalling – can help treat this neurodegeneration in mice. Neurodegenerative diseases are also often linked to problems with the cellular structures called mitochondria that provide the cell with energy in the form of the chemical ATP. Previous research suggests that abnormal mitochondrial activity and energy deficiency could be a critical step that leads to neuron death in neurodegeneration. So far, the effect of rapamycin on energy deficiency in neurons has not been explored in detail.

Zheng, Boyer et al. have now tested the therapeutic potential of rapamycin in a genetic disease called maternally inherited Leigh syndrome in which children suffer from severe neurodegeneration due to defects in their mitochondria. The experiments made use of neurons that could be grown in the laboratory and which faithfully mimicked the problems observed in maternally inherited Leigh syndrome patients. In some experiments, healthy neurons were treated with chemicals that inhibit ATP production. In other experiments, cells collected from a maternally inherited Leigh syndrome patient were coaxed into becoming neurons. Signaling via mTOR was enhanced in both kinds of neurons. Zheng, Boyer et al. then treated the defective neurons with rapamycin, which led to a significant rise in ATP levels. The production of proteins also slowed down. This could explain the observed rise in ATP levels, as making proteins consumes a lot of energy.

Zheng, Boyer et al. propose that a mild reduction in protein synthesis may have the potential to treat neurodegeneration caused by defective mitochondria. Further work is needed to extend this analysis to animal models of neurodegenerative diseases.

mouse model of Alzheimer's disease, deleting one mTOR allele decreases amyloid-β deposits and ameliorates memory deficits possibly through enhanced autophagy (*Caccamo et al., 2014*).

Mitochondrial dysfunctions are frequently observed in neurodegenerative diseases (*Lin and Beal, 2006*). Proteins causing neuronal degeneration often have either direct or indirect deleterious effects on mitochondrial functions. For example, α-synuclein inhibits mitochondrial fusion and causes mitochondrial fragmentation followed by a decrease in mitochondrial respiration and neuronal death (*Kamp et al., 2010*; *Nakamura et al., 2011*). Pink1 and Parkin are critical in mitochondrial quality maintenance (*Pickrell and Youle, 2015*). Amyloid precursor proteins accumulate in mitochondrial import channels, resulting in mitochondrial dysfunction as a hallmark of Alzheimer's disease pathology (*Devi et al., 2006*). In Huntington's disease, mutant huntingtin also has detrimental effects on mitochondrial function (*Zuccato et al., 2010*). Thus, it appears that mitochondrial dysfunction and bioenergetic collapse could be a critical step towards neuronal death.

Mitochondrial dysfunction results in decreased ATP levels in neurons. The delicate influence of ATP level on neuronal survival is best exemplified by maternally inherited Leigh syndrome (MILS), a mitochondrial DNA (mtDNA) disease, characterized by severe early childhood neurodegeneration. T8993G in *MT-ATP6*, encoding an ATP synthase subunit, is the most common mutation in MILS (*Finsterer, 2008*). A unique feature of mtDNA disease is that disease severity is correlated with the mutation load, i.e., the percentage of mutated mitochondrial DNA copies (*Taylor and Turnbull, 2005*). Higher than 90% *ATP6* T8993G causes MILS, whereas, 70~90% causes a less severe disease called NARP syndrome with symptoms, such as neuropathy, ataxia, and retinitis pigmentosa, that gradually develop with age. In a cybrid study where patient platelets containing the T8993G mtDNA mutation were fused to human osteosarcoma cells devoid of mtDNA, ATP synthesis was found to be

negatively correlated with the mutation load (*Mattiazzi et al., 2004*), indicating that a moderate difference in ATP level can dictate disease severity and the extent of neuronal death.

mTOR inhibition by rapamycin greatly attenuates neurodegeneration caused by mitochondrial complex I defects (*Johnson et al., 2013b*). This study showed a dramatic therapeutic effect of rapamycin on a mouse model of Leigh syndrome, deficient in *Ndufs4*, a nuclearly-encoded component of complex I. The life span is significantly extended, and neuronal degeneration is greatly attenuated. The exact rescue mechanism is unclear, but autophagy or mitochondrial biogenesis was excluded. It is not known if mTORC1 inhibition by rapamycin would have similar beneficial effects on the mutations affecting other respiratory complexes (*Vafai and Mootha, 2013*).

So far, rapamycin's effects on neuronal bioenergetics have not yet been explored. Here, we show that rapamycin significantly preserves neuronal ATP levels, particularly when mitochondrial oxidative phosphorylation is impaired by mitochondrial inhibitors. To test the therapeutic potential of rapamycin on neurodegeneration due to energy deficiency, we developed an iPSC-based disease model of maternally inherited Leigh syndrome (MILS), due to a T8993G mtDNA mutation in the *ATP6* gene. The MILS neurons exhibited energy defects and degenerative phenotypes consistent with patient clinical observations. Rapamycin treatment significantly alleviated ATP deficiency, reduced aberrant AMPK activation in MILS neurons and improved their resistance to glutamate toxicity. Mechanistically, MILS neurons and neurons treated with mitochondrial inhibitors all exhibited enhanced mTORC1 activity, signified by elevated ribosomal S6 and S6 kinase phosphorylation, indicating a causal link between mitochondrial dysfunction and mTOR signaling in neurons, and providing a rationale for treatment with rapamycin, which reduces protein synthesis, a major energy-consuming process.

## Results

### Rapamycin preserves neuronal ATP level

The effect of rapamycin on cellular ATP level was examined in neurons derived from human embryonic stem cells, an approach that has been successfully used to model a variety of neurological diseases (*Qiang et al., 2013*). Three mitochondrial drugs were used to mimic mitochondrial oxidative defects: oligomycin, blocking the ATP synthase; rotenone and antimycin-A, inhibiting complexes I and III, respectively, and CCCP, a mitochondrial uncoupler. We first tested whether rapamycin would affect neuronal ATP level. After a 6 hr rapamycin treatment of cultured wild type neurons differentiated from human neuroprogenitor cells (NPCs) derived from H9 human ESCs, the ATP level was increased by ~13% compared to neurons treated with DMSO as control. FK-506 (tacrolimus) that binds FKBP12, which is also a rapamycin target protein, but inhibits calcineurin signaling rather than the mTOR pathway (*Taylor et al., 2005*), did not change the ATP level (*Figure 1A*). Oligomycin treatment alone decreased neuronal ATP level to ~64% of that in neurons treated with DMSO, but strikingly, cotreatment with oligomycin plus rapamycin maintained the ATP level at ~86% (*Figure 1A*). Consistent with the higher ATP level, neurons cotreated with rapamycin showed lower AMPK T172 phosphorylation, an indicator of cellular ATP deficiency, compared to treatment with oligomycin alone (*Figure 1B*). Similar effects of rapamycin were observed in neurons treated with rotenone and antimycin-A; but, interestingly, rapamycin was not able to preserve ATP when neurons were treated with CCCP (*Figure 1A*). It should be noted that both oligomycin and rotenone/antimycin-A treatment reduce ATP production by directly inhibiting oxidative phosphorylation; in contrast, CCCP does so by uncoupling electron transport from ATP production, which not only reduces ATP production, but also stimulates oxidative phosphorylation and induces mitochondrial substrate burning and heat production. We suspect that this difference may account for the different effects of cotreatment with rapamycin. These data indicate that rapamycin can increase neuronal ATP levels and preserve cellular energy when oxidative phosphorylation is impaired.

### Increased ribosomal S6 and S6 kinase phosphorylation in neurons treated with mitochondrial OXPHOS inhibitors

Phosphorylation of ribosomal protein S6, a target of mTOR complex 1 (mTORC1) signaling, is increased in the brain lysate of *Ndufs4* -/- mice, although it is unknown in what type of brain cells, i.e. neurons or glial cells, this occurs (*Johnson et al., 2013b*). We found an ~2-fold increase in

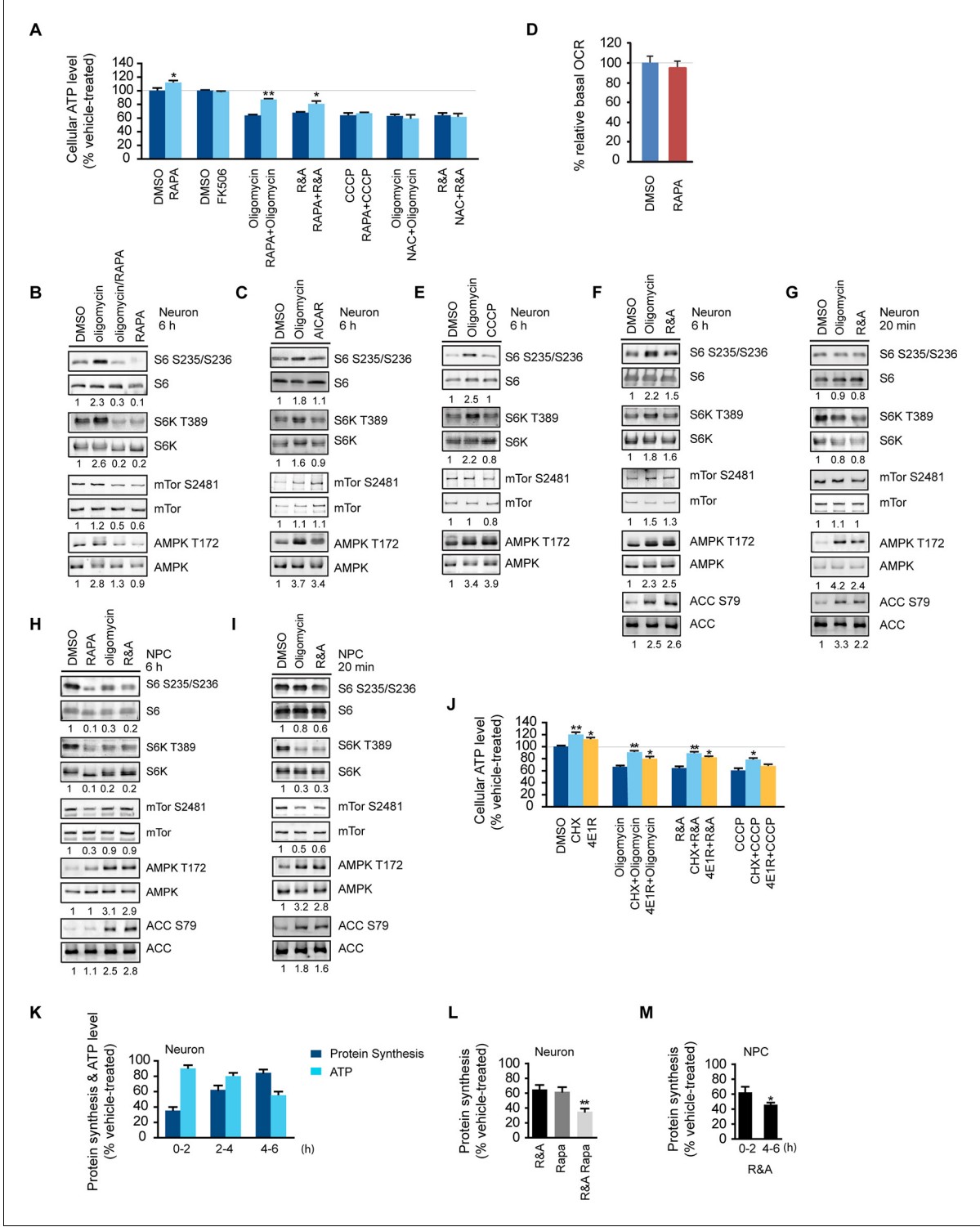

**Figure 1.** Rapamycin treatment increased neuronal ATP levels. (**A**) The effect of rapamycin (RAPA) on cellular ATP level was examined in 5-week neurons differentiated from human neuroprogenitor cells (NPCs) derived from H9 ESCs. Rapamycin was used at 20 nM (final concentration). Mitochondrial dysfunction was mimicked by chemicals disrupting mitochondrial oxidative function: oligomycin (2 μM), blocking complex V (ATP synthase); rotenone and antimycin A (R&A; 1 μM each), complex I and III inhibitors; CCCP (20 μM), a mitochondrial uncoupler. All were prepared in DMSO as vehicle. N-acetylcysteine (NAC) was used at 750 μM (final concentration). The treatment was done for 6 hr with neurons grown in duplicate wells from the same batch of differentiation. The relative ATP level for each treatment was calculated as percentage after normalization to DMSO-treated neurons. Bars are mean ± SD, n=3. *p<0.05. **p<0.01, calculated by two-tailed t-test. (**B**) Immunoblot analysis of cell lysates prepared from

*Figure 1 continued on next page*

*Figure 1 continued*

neurons treated with oligomycin, rapamycin or both for 6 hr. The intensity of phosphorylated protein was quantified after normalization to non-phosphorylated signal, and was presented as fold change compared to control group treated with DMSO. (C) Immunoblot analysis of cell lysates prepared from neurons treated with oligomycin or AICAR for 6 hr. (D) Oxygen consumption rate (OCR) measurement by Seahorse extracellular flux analyzer. The basal OCRs of neurons treated with rapamycin for 6 hr were compared to neurons treated with DMSO as control. Bars are mean ± SD, n=3. (E) Immunoblot analysis of cell lysates prepared from neurons treated with CCCP for 6 hr. (F) Immunoblot analysis of cell lysates prepared from neurons treated with rotenone and antimycin-A for 6 hr. (G) Immunoblot analysis of cell lysates prepared from neurons treated with oligomycin or rotenone & antimycin-A for 20 min. (H) Immunoblot analysis of cell lysates prepared from NPCs treated with rapamycin, oligomycin, and rotenone/antimycin-A for 6 hr. (I) Immunoblot analysis of cell lysates prepared from NPCs treated with oligomycin or rotenone & antimycin-A for 20 min. (J) The effect of protein synthesis inhibition on cellular ATP level was examined in 5-week neurons differentiated from human neuroprogenitor cells (NPCs) derived from H9 ESCs. Cycloheximide (CHX) was used at 20 μg/ml, and 4E1RCat was used at 50 μM. The treatment was done for 2 hr with CHX and 4E1RCat alone, and for 6 hr when combined with mitochondrial inhibitors with neurons grown in duplicate wells from the same batch of differentiation. (K) Five-week neurons differentiated from human neuroprogenitor cells (NPCs) derived from H9 ESCs were treated with vehicle (DMSO) or rotenone & antimycin-A (R&A). Protein synthesis was measured by pulsing for 2 hr with $^{35}$S-Cys/Met every 2 hr from 0 to 6 hr, and $^{35}$S incorporation into protein and neuronal ATP levels were quantified and normalized to the DMSO-treated controls. Data are mean ± SD, n=3. (L) Five-week neurons differentiated from human neuroprogenitor cells (NPCs) derived from H9 ESCs were treated for vehicle (DMSO), rotenone & antimycin-A, rapamycin or both (R&A Rapa) for 4 hr. Protein synthesis was measured by labeling for 2 hr with $^{35}$S-Cys/Met from 2 to 4 hr. **p<0.01, calculated by two-tailed t-test. (M) Protein synthesis in NPCs derived from H9 ESC treated with rotenone & antimycin-A for 6 hr. Data are mean ± SD, n=3. All the experiments were repeated at least three times. (see associated *Figure 1—source data 1*).

The following source data and figure supplements are available for figure 1:

**Source data 1.**

**Figure supplement 1.** Five-week neurons differentiated from human neuroprogenitor cells (NPCs) derived from H9 ESCs were treated for vehicle (DMSO), cycloheximide and 4E1RCat.

**Figure supplement 2.** The glucose concentration in the medium growing 3-week neurons derived from H9 ESCs treated with DMSO and rapamycin for 8 hr were quantified by YSI 2950 metabolite analyzer.

ribosomal S6 and S6 kinase phosphorylation in neurons treated for 6 hr with oligomycin or rotenone/antimycin-A, but not CCCP (*Figure 1B,E and 1F*). Rapamycin only partially decreased mTOR S2481 phosphorylation as previously reported (*Hsu et al., 2011*), but almost completely abolished the ribosomal S6 and S6K phosphorylation observed in oligomycin-treated neurons, indicating its dependence on mTORC1 (*Figure 1B*). We did not observe a consistent change of mTOR phosphorylation at S2481 or S2448 (not shown) upon oligomycin or rotenone/antimycin-A treatment. The increased S6 and S6 kinase phosphorylation was not due to AMPK activation, as AICAR, an AMPK agonist, did not alter their phosphorylation (*Figure 1C*). In fact, AMPK activation is generally associated with decreased mTORC1 activity and S6K phosphorylation as a result of direct phosphorylation of Tsc2 and Raptor by AMPK (*Inoki et al., 2003*; *Gwinn et al, 2008*). Consistent with this, NPCs treated with oligomycin or rotenone/antimycin-A for 20 min or for 6 hr showed decreased S6 and S6K phosphorylation, increased phosphorylation of AMPK and its substrate, acetyl-CoA carboxylase (ACC) (*Figure 1H and 1I*). In contrast to the 6 hr treatment, 20 min of oligomycin or rotenone/antimycin-A treatment did not significantly alter S6 and S6K phosphorylation in neurons (*Figure 1G*). Taken together, these results suggest that the increased S6 and S6K phosphorylation in neurons treated for 6 hr with these mitochondria inhibitors is a neuron-selective response to some cumulative effect caused by mitochondrial dysfunction.

## Protein synthesis inhibition spares a significant amount of ATP in neurons

As rapamycin was able to preserve ATP in neurons treated with rotenone/antimycin-A, which largely abolishes oxidative phosphorylation, it is unlikely that the effect of rapamycin is through increasing ATP production from mitochondrial oxidative phosphorylation. Nevertheless, we measured the basal oxygen consumption rate in neurons treated with rapamycin, and the rate was similar to that in DMSO-treated controls, supporting the conclusion that rapamycin's effect does not come from increasing oxidative phosphorylation activity (*Figure 1D*). Rapamycin treatment did not increase

neuronal glucose consumption indicating that glycolysis was not increased (*Figure 1—figure supplement 2*). Mitochondrial dysfunction leads to increased production of reactive oxygen species causing cellular damage; however, N-acetylcysteine (NAC), a commonly used anti-oxidant, was unable to maintain the ATP level in this assay (*Figure 1A*).

With long-term treatment, both oligomycin and rotenone/antimycin-A treated neurons showed increased ribosomal S6 and S6K phosphorylation, which are indicators of mTORC1 signaling and, indirectly, the rate of protein synthesis, a major cellular energy-consuming process (*Rolfe and Brown, 1997*), and the increase in ribosomal S6 phosphorylation in oligomycin-treated neurons was almost completely abolished by rapamycin treatment, which inhibits mTORC1 (*Figure 1B*). One consequence of rapamycin inhibition of mTORC1 is a decrease in protein synthesis, and to explore the potential ATP-saving effect of protein synthesis inhibition, neurons were treated for 2 hr with cyclo-heximide (CHX), a widely-used protein synthesis inhibitor, and 4E1RCat, a protein synthesis initiation inhibitor that blocks eIF4E:eIF4G and eIF4E:4E-BP1 interactions (*Cencic et al., 2011*), causing a drop in protein synthesis to ~9% and ~55% of that of neurons treated with DMSO as control, respectively (*Figure 1—figure supplement 1*). As a result of this inhibition, the neuronal ATP level increased by ~26% and ~14%, respectively (*Figure 1J*). Similar to rapamycin, cycloheximide and 4E1RCat could also significantly preserve ATP levels in neurons treated with mitochondrial inhibitors (*Figure 1J*). These results imply that reduction of protein synthesis is an important factor for rapamycin to preserve ATP in neurons.

The increased S6 and S6K phosphorylation in neurons treated with mitochondrial inhibitors at later times implies that an increase in protein synthesis should occur. To monitor neuronal protein synthesis during the treatment with mitochondrial inhibitors, we labeled neuronal cultures with $^{35}$S-cysteine/methionine at different time points after rotenone/antimycin-A treatment. During the first 2 hr, protein synthesis dropped to ~36% of that of neurons treated with DMSO as control, but, interestingly, protein synthesis recovered and reached ~80% by 4–6 hr despite the sustained decrease in neuronal ATP levels (*Figure 1K*). Such increased protein synthesis would consume more ATP and presumably aggravate the energy crisis. When rapamycin was added together with rotenone/antimycin-A, after 4 hr treatment, protein synthesis was ~38% of the control, while in neurons treated with rotenone and antimycin-A, the protein synthesis rate was ~62%, and rapamycin treatment alone decreased protein synthesis to ~60% (*Figure 1L*). Therefore, rapamycin may help to preserve ATP through reducing protein synthesis. Protein synthesis was not restored in NPCs treated with rotenone/antimycin-A for 6 hr (*Figure 1M*), consistent with the phosphorylation status of S6K and S6 (*Figure 1H*).

## iPSC-based model of maternally inherited Leigh syndrome

Our results so far indicated that rapamycin can preserve the ATP level in neurons treated with mitochondrial oxidative phosphorylation inhibitors, which cause acute mitochondrial dysfunction. To test the therapeutic potential of rapamycin for neurodegeneration resulting from energy deficiency, we developed an induced pluripotent stem (iPS) cell model of maternally-inherited Leigh syndrome (MILS), an infantile neurodegenerative disease due to mitochondrial DNA mutation. We obtained a clone of primary fibroblasts (GM13411) derived from a male MILS syndrome patient, who died at 8 months; the patient's symptoms, disease development and brain pathology were typical of MILS syndrome as described in a clinical report (*Pastores et al., 1994*). The GM13411 MILS patient fibroblast line has a T8993G mutation resulting in a change from a conserved leucine to arginine at amino acid position 156 in ATP6, a subunit of Complex V/ATP synthase. Three iPS cell lines (iPSCs) were established from GM13411 fibroblasts using a standard cocktail of reprogramming retroviruses, expressing *OCT4 (POU5F1), SOX2, KLF4*, and *MYC* (*Takahashi et al., 2007*). Pluripotency markers were assessed by RT-PCR (*Figure 2—figure supplement 1*) and immunostaining (*Figure 2A*). Healthy control iPS cell lines were derived from BJ male human fibroblasts. Both the T8993G and BJ iPSCs had normal karyotypes (*Figure 2—figure supplement 2*). The mitochondrial genomes from these T8993G and BJ iPS cells were sequenced, and no major pathogenic mutations, apart from T8993G, were found compared to mitochondrial genome variation databases (*Figure 2—figure supplement 3*). Subsequently, three neural progenitor cell lines (NPC) were derived from the respective patient iPS cell lines using the embryoid-body based protocol outlined in *Figure 2—figure supplement 4*. Expected neural progenitor markers were present by immunostaining (*Figure 2A*), and all the T8993G NPC lines retained the T8993G mutation (*Figure 2B* and *Figure 2—figure supplement*

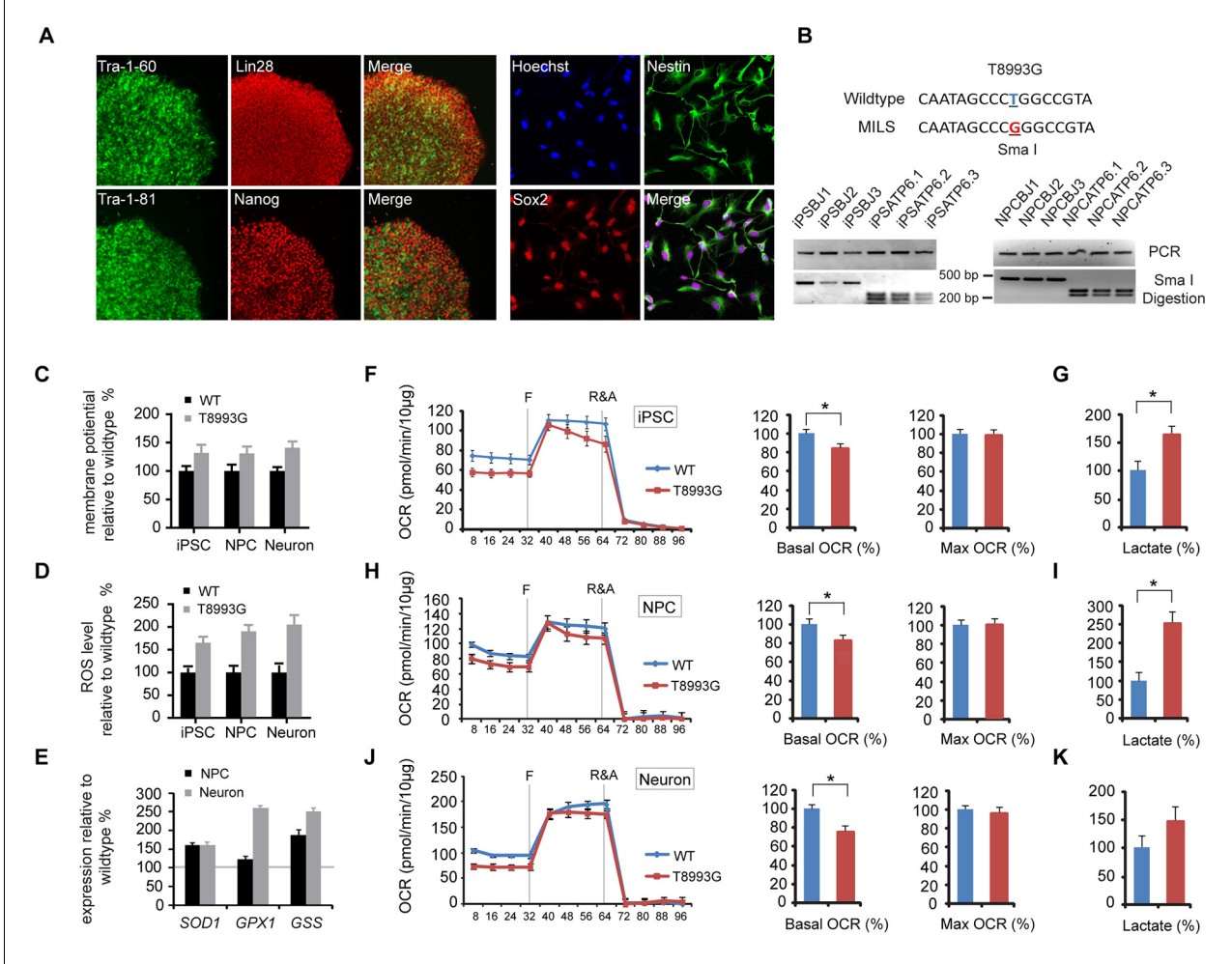

**Figure 2.** Established iPSCs and neuroprogenitor cells (NPC) from GM13411, a MILS fibroblast line. (**A**) T8993G iPSC expressed pluripotency markers that included Tra-1–60, Lin28, Tra-1–81 and Nanog. NPCs derived from T8993G iPSCs were stained with anti-Sox2 and Nestin. (**B**) T8993G mutation generates a Sma I restriction enzyme site. T8993G iPSCs and NPC cells still retained the mutation as confirmed by PCR and Sma I digestion. DNA products were separated on agarose gel by electrophoresis. (**C**) Mitochondrial membrane potential analyzed by fluorescence-activated cell sorting (FACS) using TMRE staining. Two lines of iPSCs, NPCs and neurons derived from BJ fibroblasts and one from H9 hESCs were used as controls (WT). The relative mitochondrial membrane potential was presented as percentage compared to the mean of control. Bars are mean ± SD, n=3. The experiment was repeated three times. (**D**) Cellular reactive oxygen species (ROS) analyzed by FACS using CM-H2DCFDA staining. Two lines of iPSCs, NPCs and neurons derived from BJ fibroblasts and one from H9 hESCs were used as control (WT). The relative ROS level was presented as percentage compared to the mean of control. Bars are mean ± SD, n=3. The experiment was repeated three times. (**E**) T8993G NPCs and neurons had higher expression of oxidative stress response genes including *SOD1, GPX1* and *GSS*. Two lines of iPSCs, NPCs and neurons derived from BJ fibroblasts and one from H9 hESCs were used as control (WT). The gene expression levels were quantified by real-time PCR after normalization to β-actin. The relative expression level was presented as percentage compared to the mean of control. Bars are mean ± SD, n=3. The experiment was repeated three times. (**F, H, J**) Oxygen consumption rate (OCR) measured by Seahorse extracellular flux analyzer. FCCP (**F**) is a mitochondrial uncoupler; rotenone and antimycin A (R&A) are complex I and III inhibitors. Error bars represent SD, n=6. Non-mitochondrial oxygen consumption has been subtracted. The relative percentage of basal and maximum OCR of T8993G iPSC, NPC and neurons at 3 weeks of differentiation were calculated by comparing to the mean of BJ and H9 cells (WT). The original data was in ***Figure 2—figure supplement 9***. (**G, I, K**) Measurement of lactate secreted by iPSCs, NPCs and neurons at 3 weeks of differentiation. The relative percentage of secreted lactate from T8993G iPSC, NPC and neurons was calculated by comparing to the mean of BJ and H9 cells. Bars represent mean ± SD. n=3. *p<0.05. Calculated by two-tailed t-test. The experiments were repeated three times. (see associated ***Figure 2—source data 1***).

The following source data and figure supplements are available for figure 2:

**Source data 1.**

**Figure supplement 1.** RT-PCR analysis of pluripotency genes, *OCT4, NANOG*, KLF4 and *SOX2* in T8993G and BJ iPSCs.

*Figure 2 continued on next page*

*Figure 2 continued*

**Figure supplement 2.** Karyotypes of three T8993G iPSC clones, 46, XY; and one BJ iPSC, 46, XY.

**Figure supplement 3.** Sequencing of mitochondrial DNA extracted from T8993G (GM13411) and BJ iPSCs.

**Figure supplement 4.** The outline of the protocol used to differentiate neurons from iPSCs; representative pictures of fibroblasts, iPSCs, embryoid bodies (EB) and neural rosettes.

**Figure supplement 5.** Sanger sequencing confirmed the T8993G mutation (upper panel, representative result).

**Figure supplement 6.** T8993G iPSCs, NPC cells and neurons all had an extremely high T8993G mtDNA mutation load as GM13411 fibroblast.

**Figure supplement 7.** Neuronal marker staining.

**Figure supplement 8.** Electrophysiological study of T8993G and BJ 5-week neurons.

**Figure supplement 9.** Oxygen consumption rate (OCR) analysis by Seahorse extracellular flux analyzer on BJ, H9 and *ATP6* T8993G iPSCs, NPCs and 3-week neurons.

*5*). Like the GM13411 fibroblasts, the derived iPSCs, NPCs and neurons all had an extremely high T8993G mtDNA mutation load (*Figure 2—figure supplement 6*). The staining of neuronal differentiation markers and electrophysiological analysis of patient and BJ neurons is described in supplementary data; representative results for T8993G neurons were shown in *Figure 2—figure supplement 7*, *8*.

## T8993G MILS cells recapitulate the mitochondrial defects found in cybrid studies

T8993G iPSCs, NPCs and neurons all exhibited increased mitochondrial membrane potential and cellular ROS level compared to the control BJ- and H9-derived lines (*Figure 2C and 2D*). ROS responsive genes, such as *SOD1*, *GPX1* and *GSS*, were also up-regulated in T8993G NPCs and neurons (*Figure 2E*). These data are consistent with the previous studies based on a cybrid model, in which the *ATP6* T8993G mutation was found to impair the proton passage through ATP synthase, which leads to higher mitochondrial membrane potential and ROS production (*Cortés-Hernández et al., 2007*; *Manfredi et al., 1999*; *Mattiazzi et al., 2004*; *Trounce et al., 1994*). Extracellular flux analysis of oxygen consumption rate (OCR) revealed that iPSCs, NPCs and neurons containing T8993G mtDNA had a lower basal OCR but similar maximum OCR to controls (*Figure 2F,H and 2J*), mechanistically consistent with ATP synthase deficiency. Significantly more lactate was secreted by T8993G iPSCs and NPCs than controls (*Figure 2G and 2I*), whereas T8993G neurons exhibited only a small increase in secreted lactate (*Figure 2K*). This result indicates enhanced aerobic glycolysis in proliferating T8993G cells. Recently, *Ma et al. (2015)* reported the establishment of iPSCs from the same patient fibroblast line, which have metabolic phenotypes similar to our T8993G iPSC lines. Although they did not differentiate patient neurons from iPSCs, by using somatic cell nuclear transfer (SCNT) technology, they replaced the mutant mtDNA and generated corrected pluripotent stem cells, which had a normal metabolic profile, proving that the metabolic phenotypes observed in the patient iPSCs are due to mtDNA mutation.

## Shutoff of aerobic glycolysis during neuronal differentiation exposes ATP synthesis deficiency in T8993G MILS neurons

Interestingly, only T8993G neurons showed a significant ATP shortage; in contrast, T8993G iPSCs and NPCs had slightly lower but comparable ATP levels to controls (*Figure 3A*). The ATP level in MILS neurons dropped to ~73% of healthy control neurons, and, consistently, phosphorylation of AMPK T172, an indicator of ATP shortage, and its substrate ACC was significantly increased in T8993G neurons but not in T8993G NPCs, iPSCs and fibroblasts (*Figure 3A*). Similarly, oligomycin,

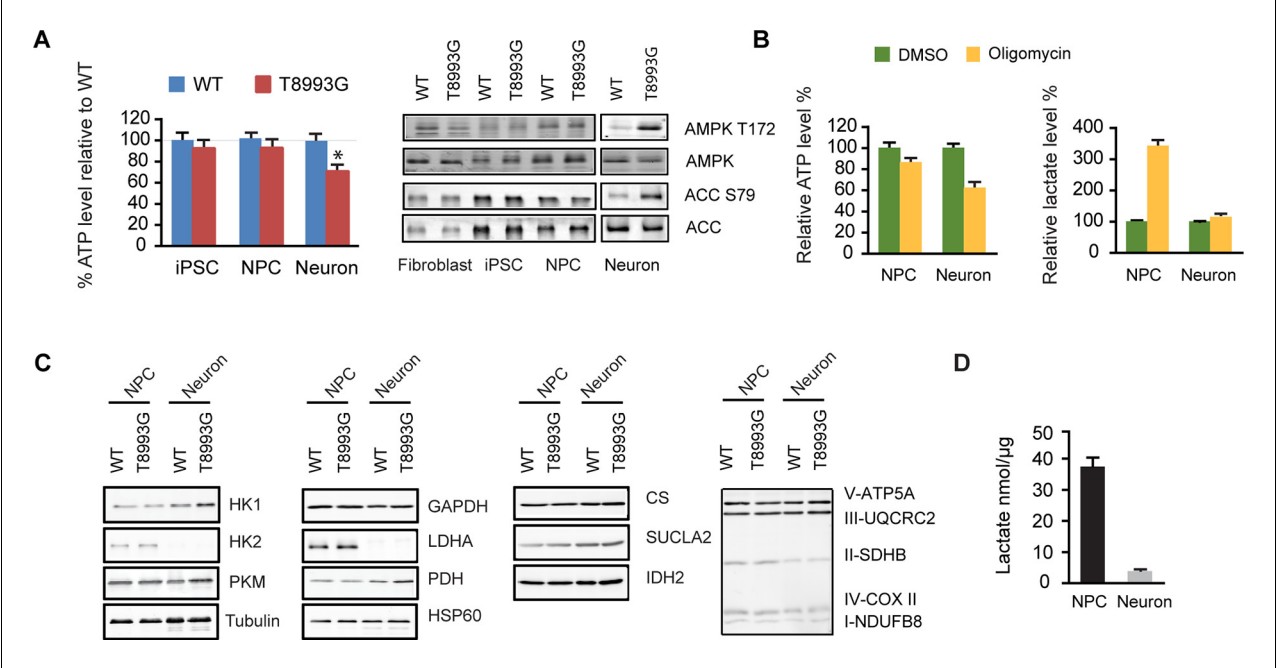

**Figure 3.** Shutoff of aerobic glycolysis during neuronal differentiation exposes mitochondrial ATP synthesis deficiency in T8993G MILS neurons. (**A**) Relative ATP level of T8993G compared to healthy control (BJ, H9 hESC) in iPSCs, NPCs and neurons. The relative percentage of ATP levels in T8993G was calculated by comparing to the mean of control cells respectively. Bars are mean ± SD, n=3. *p<0.05. Calculated by two-tailed t-test. Immunoblot analysis of AMPK Thr172 and ACC Ser79 phosphorylation in cell lysates prepared from primary fibroblasts, iPSCs, NPCs and neurons. (**B**) Cellular ATP level and secreted lactate from H9 NPCs and neurons treated with DMSO and oligomycin for 6 hr. The relative percentage of ATP levels was calculated by comparing to the mean of DMSO-treated cells respectively. Bars are mean ± SD, n=3. (**C**) Immunoblot analysis of representative enzymes in glycolysis, TCA and mitochondrial respiratory complexes in BJ and T8993G NPCs and neurons. 20 μg protein lysate from each sample were loaded for SDS-PAGE. (**D**) Measurement of lactate secreted by NPCs and neurons derived from human BJ iPSCs at 3 weeks. NPC and differentiated neurons at 3 weeks were incubated in fresh medium for 12 hr, and lactate in the medium is quantified. Bars represent mean ± SD of the absolute concentration of lactate after normalized to protein content. n=3. All the experiments were repeated at least three times. (see associated *Figure 3—source data 1*).

The following source data is available for figure 3:

**Source data 1.**

an ATP synthase inhibitor, dramatically reduced ATP levels in wild-type neurons but less significantly in NPCs; and lactate levels increased significantly in NPCs but not neurons (*Figure 3B*). These results are consistent with the fact that neurons mainly rely on mitochondria for energy production. Immunoblot analysis of representative enzymes in the glycolysis and TCA pathways, and mitochondrial respiratory complexes showed no major differences between T8993G and BJ control NPCs or neurons (*Figure 3C*). Strikingly, however, neurons did not express detectable hexokinase (HK2) or lactate dehydrogenase (LDHA) proteins, the two key enzymes supporting aerobic glycolysis (*DeBerardinis and Thompson, 2012*; *Dang, 2012*). Consistently, the production of lactate in wild type neurons was ~10 fold less than in NPCs (*Figure 3D*). Presumably, in iPSCs and NPCs, the increased production of lactate allows more NADH to be recycled to $NAD^+$, which is required for the conversion of glyceraldehyde 3-phosphate into 1,3-bisphosphoglycerate, and results in production of more glycolytic ATP. Without LDHA and HK2, neurons appear unable to compensate for the mitochondrial ATP deficiency through aerobic glycolysis. Moreover, the data also argue that even when short of energy, neurons cannot turn on aerobic glycolysis, at least in MILS neurons. In spite of ATP deficiency in T8993G neurons, we did not detect enhanced autophagy (data not shown), probably due to mTOR activation, which suppresses autophagy. Moreover, there was no marked increase in mitochondrial mass in T8993G neurons (*Figure 3C*), indicating that mitochondrial biogenesis was not deployed to compensate for cellular energy deficiency.

## Degenerative phenotype of T8993G MILS neurons

T8993G neurons showed finer neuronal fibers than control neurons, and bead-like structures along the T8993G axons were more common (*Figure 4A and 4B*). Neurons at 3 and 8-weeks of differentiation retained the homoplasmic T8993G mutation of the parental GM13411 fibroblasts (*Figure 4C*). The bead-like structures, known as neuritic beading, are focal swellings in the axons and dendrites of neurons, which occur upon intracellular ATP decrease (*Takeuchi et al., 2005*). Mitochondrial dysfunction increases neuronal vulnerability to form neuritic beading (*Greenwood et al., 2007*). To challenge the MILS neurons with energy demanding stress, we used a glutamate toxicity assay. Glutamate overdosage is toxic to neurons through two mechanisms - excessive stimulation of neuronal activity and non-receptor oxidative toxicity (*Rothman, 1985*; *Murphy et al., 1989*). ATP deficiency is the primary trigger for neuronal toxicity caused by glutamate overdosage (*Nicholls et al., 2007*). T8993G neurons were hypersensitive to increased levels of extracellular glutamate (*Figure 4D and 4E*). In neurons expressing GFP driven by the neuron-specific DCX promoter to highlight neuronal structures, a 6-h treatment with 100 µM glutamate abolished neuronal processes in T8993G neurons, whereas the neuronal fibers from control neurons, differentiated from BJ iPSCs or H9 hESCs, remained intact, and could tolerate up to 300 µM glutamate (*Figure 4D and 4H*). After a 3-h treatment, neuronal fibers, containing multiple neuritic beadings, were already apparent in T8993G neurons (*Figure 4F*). Consistently, neuronal ATP levels dropped during glutamate treatment (*Figure 4G*). To confirm this observation, we used oligomycin, an ATP synthase inhibitor to mimic the defect caused by the *ATP6* T8993G mutation. By measuring the basal OCRs with different concentration of oligomycin, 40 nM was found to partially inhibit ATP synthase (*Figure 4—figure supplement 1*). Wild type neurons treated with 40 nM oligomycin became sensitive to 100 µM glutamate similar to MILS neurons (*Figure 4H and 4I*), and neuronal ATP levels fell consistent with their sensitivities to glutamate (*Figure 4J*).

Besides low ATP itself, AMPK activation may also contribute to the deleterious effect of the T8993G mutation. We found that AICAR treatment to activate AMPK from the start of neuronal differentiation led to collapse of neuronal extensions (*Figure 4—figure supplement 2*), while treatment of already differentiated neurons with AICAR led to significant cell death (*Figure 4—figure supplement 3*). This is consistent with the finding that activation of AMPK suppresses axon initiation and neuronal polarization by phosphorylation of KIF5, the motor protein for the kinesin light chain (*Amato et al., 2011*). However, it should be noted that the intensity of AMPK T172 phosphorylation triggered by AICAR was stronger in BJ and H9 neurons than in T8993G neurons (not shown).

## Rapamycin treatment significantly increases ATP level and decreases aberrant AMPK phosphorylation in T8993G MILS neurons

The *ATP6* T8993G mutation, similar to oligomycin treatment, hampers proton intake through ATP synthase. Like oligomycin-treated healthy control neurons, neurons differentiated from independent T8993G NPC lines all showed significantly increased ribosomal S6 and S6K phosphorylation compared to BJ and H9, while T8993G NPCs did not show increased S6 and S6K phosphorylation (*Figure 5A*). Six-h rapamycin treatment of T8993G neurons increased the ATP level by ~23% compared to those treated with DMSO as control; and consistently, AMPK T172 phosphorylation also decreased (*Figure 5B*). In spite of significantly lower cellular ATP levels and activated AMPK, the rate of protein synthesis in T8993G neurons was still about 92% of BJ control neurons (*Figure 5C*). Rapamycin treatment also helped T8993G neurons cope with the stress of glutamate treatment; the neural fibers of T8993G neurons treated with rapamycin sustained a 6 hr 100 µM glutamate treatment (*Figure 5D*).

To further confirm the therapeutic effect of rapamycin, we established another model of ATP synthase deficiency using shRNA-mediated knockdown of ATP5A1, a nuclearly-encoded key component of ATP synthase. ATP5A1 deficiency is also known to cause fatal neonatal mitochondrial encephalopathy (*Jonckheere et al., 2013*). Consistent with our previous observations, ATP5A1-depleted neurons but not NPCs showed increased S6 and S6K phosphorylation similar to T8993G cells (*Figure 5E*). ATP5A1-depleted neurons at 3-weeks of differentiation already showed a significant number of beaded neuronal processes compared to neurons infected with scramble control shRNA, and the ATP level of ATP5A-depeleted neurons was only ~60% of the neurons transduced with scramble control shRNA (*Figure 5F*). Twelve-hr treatment of rapamycin increased ATP level by

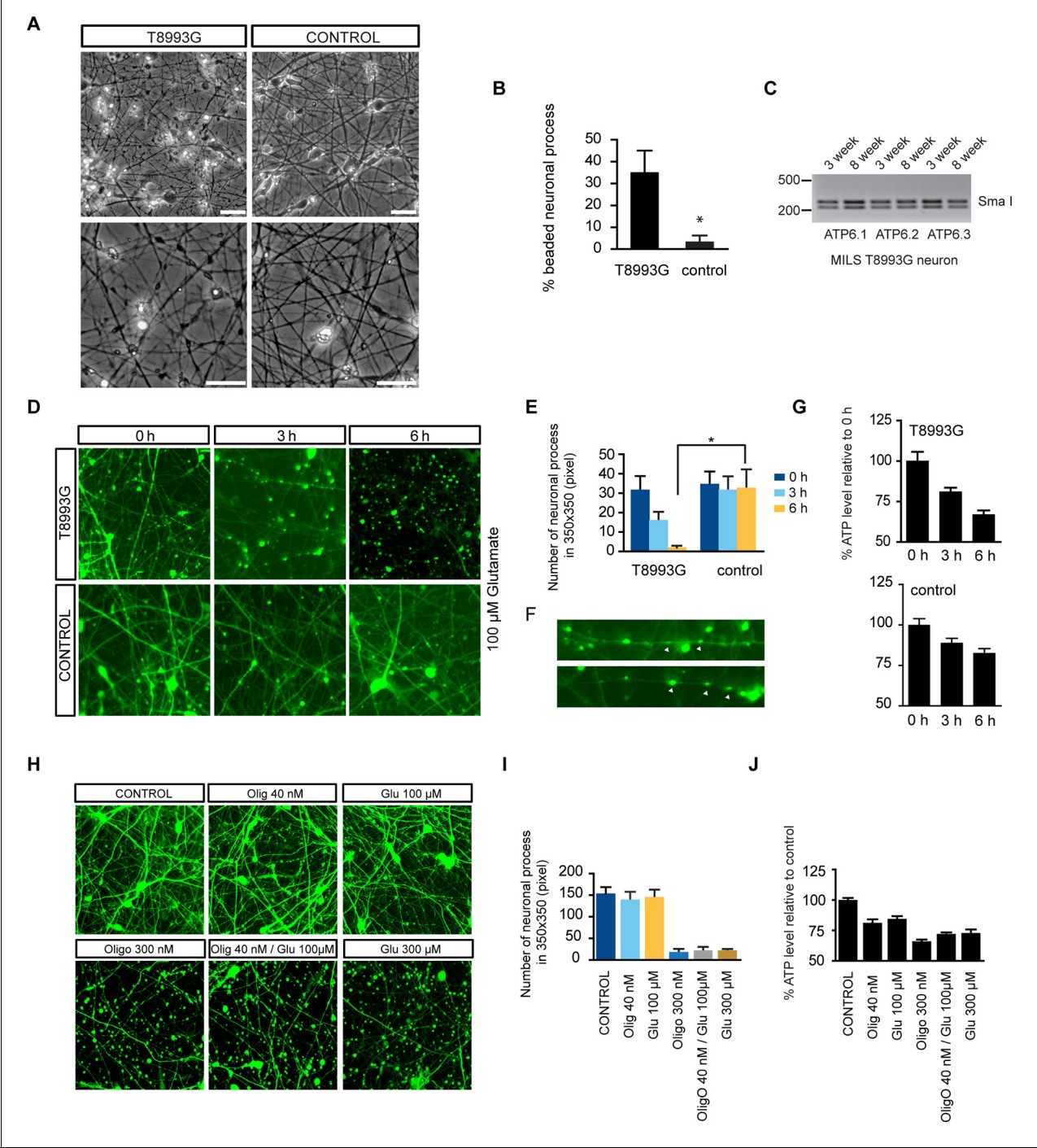

**Figure 4.** Degenerative phenotype of T8993G MILS neurons. (**A**) Phase contrast photo of T8993G and BJ neurons at 8 weeks of differentiation. Scale bar, 20μm. (**B**) The percentage of neuronal processes containing neuritic beads was quantified by counting 20 neuronal processes for T8993G, BJ and H9 neurons as control. Bars are mean ± SD, n=3. *p<0.05 calculated by two-tailed t-test. (**C**) T8993G neurons differentiated at 3 and 8 weeks still retained the original high T8893G mutation load as confirmed by PCR and Sma I digestion. DNA products were separated on an agarose gel by electrophoresis. (**D**) Glutamate-induced toxicity test. Eight-week T8993G and BJ neurons containing DCX promoter-driven GFP were treated with 100 μM glutamate in neuron growth medium. (**E**) To quantify the extent of neuronal process collapse, the number of discernible neuronal processes in a fixed photo area (350x350 pixel) were counted at time points of 0, 3, and 6 hr. Bars are mean ± SD, n=3. *p<0.05 calculated by two-tailed t-test. (**F**) Neuritic beads, indicated by white arrows, formed along the axons. (**G**) Cellular ATP of 8-week T8993G and BJ neurons treated with 100 μM glutamate. The relative percentage of ATP level was calculated by comparing to the mean of 0 hr cells respectively. Bars are mean ± SD, n=3. (**H**) BJ neurons containing DCX promoter-driven GFP were treated with oligomycin and glutamate in neuron growth medium for 6 hr. (**I**) Quantification of the extent of

*Figure 4 continued on next page*

*Figure 4 continued*

neuronal process collapse. (**J**) Measurement of ATP level. The relative percentage of ATP level was calculated by comparing to the mean of untreated BJ neurons respectively. Bars are mean ± SD, n=3. All the experiments were repeated at least three times. (see associated *Figure 4—source data 1*).

The following source data and figure supplements are available for figure 4:

**Source data 1.**
**Figure supplement 1.** Oxygen consumption rate (OCR) analysis by Seahorse extracellular flux analyzer on neurons treated with Oligomycin.
**Figure supplement 2.** Effect of AICAR on neuron differentiation.
**Figure supplement 3.** Effect of AICAR on 6-week differentiated neurons.

~26% and greatly decreased the number of beaded neuronal processes (*Figure 5G, 5H and 5I*). Cycloheximide, a protein synthesis inhibitor, also had effects comparable to rapamycin in preventing beading (*Figure 5G*). These data suggest that rapamycin treatment has the potential to benefit Leigh or NARP syndrome patients by counteracting energy deficiency.

## Elevated amino acids level in MILS neurons

To understand the mechanism through which long term treatment with oligomycin or MILS ATPase deficiency results in elevated mTORC1 activity, we defined the metabolic profiles of MILS neurons, by measuring representative glycolytic and TCA metabolites and amino acids using gas chromatography mass spectrometry (GC-MS). The levels of 19 amino acids were quantified; we were unable to measure the level of arginine due to instability. As shown in *Figure 6A*, the overall levels of amino acids in MILS neurons were increased compared to healthy control neurons: 14 amino acids showed significant increase; alanine, asparagine, histidine, isoleucine, leucine, lysine, methionine, phenylalanine, proline, serine and valine increased by ~40 to 80%; cysteine and threonine increased by ~100%; and, glycine increased by 340%. Pyruvate and lactate levels in MILS neurons were two-fold higher, and the TCA intermediates in MILS neurons were also increased; citrate was ~200% of that in control neurons, α-ketoglutarate was ~180%, and succinate was ~140%, whereas fumarate and malate were comparable to the control (*Figure 6B*). These data are consistent with the *ATP6* T8993G mutation defect, which decreases mitochondrial electron transport chain activity, resulting in decreased usage of TCA and glycolytic metabolites. The increased intracellular amino acids levels in MILS neurons may also be attributable to the accumulated glycolytic intermediates and a clogged TCA cycle, because the synthesis and catabolism of amino acids is linked to glycolysis and the TCA cycle; under catabolic conditions the carbon atoms of amino acids are oxidized by the TCA cycle for ATP production (*Stryer et al., 2002*). Since the mTORC1 complex is a sensor of cellular nutrition, and can be activated by amino acids (*Bar-Peled and Sabatini, 2014*), we suspect that the observed enhancement of neuronal mTORC1 activity after prolonged mitochondrial OXPHOS inhibition is due to the accumulated amino acids or other nutrients that activate mTORC1.

To further examine the change of these metabolites during mitochondrial inhibition, we measured metabolite levels in the neurons treated with oligomycin or rotenone and antimycin A (R&A) for 6 hr. Similar to MILS neurons, neurons treated with these mitochondrial drugs showed higher levels of amino acids, pyruvate and lactate (*Figure 6D and 6E*); although there are some notable differences between mitochondrial inhibitor-treated neurons and MILS neurons. For example, the extent of glycine increase in inhibitor-treated neurons is not as pronounced as in MILS neurons. These differences may be due to the difference in mechanism or extent of mitochondrial inhibition between mitochondrial inhibitors and *ATP6* T8993G mutation. The large increase of glycine in MILS neurons may be an anti-oxidant response to chronic mitochondrial oxidative stress. In the mitochondrial matrix, serine is converted to glycine catalyzed by serine hydroxymethyltransferase 2 (SHMT2), a reaction coupled with covalent linkage of tetrahydrofolate to a methylene group to form 5,10-methylene-tetrahydrofolate, which is subsequently used to generate 5,10-methenyl-tetrahydrofolate and NADPH by tetrahydrofolate dehydrogenase 2 (MTHFD2) in mitochondria. The NADPH produced through this serine catabolism pathway maintains mitochondrial redox by regenerating the reduced forms of

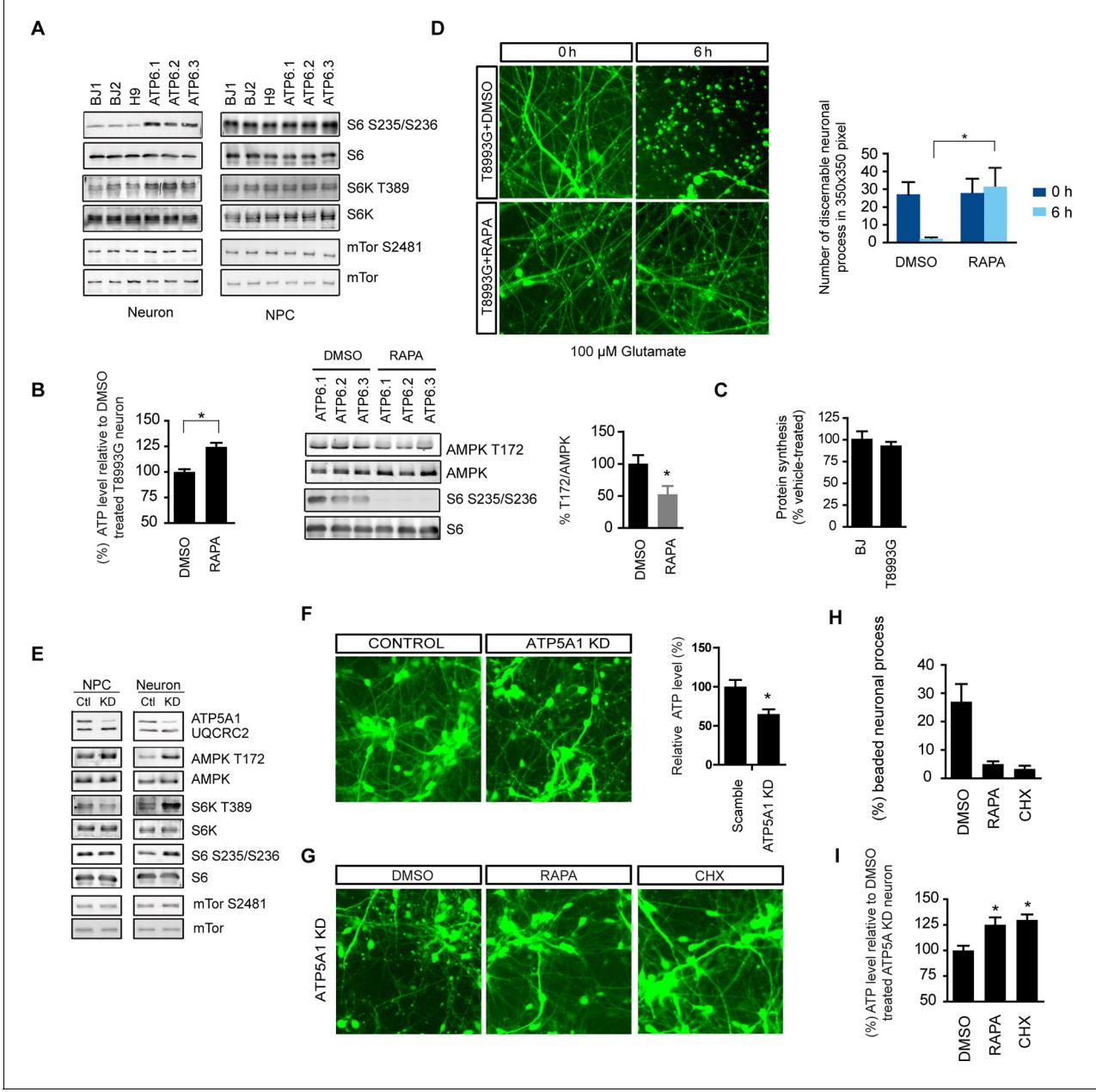

**Figure 5.** Rapamycin treatment alleviates ATP deficiency and aberrant AMPK activation in T8993G MILS neurons. (**A**) Immunoblot analysis of phosphorylation of ribosomal S6, S6K and mTOR in 3-week neurons and NPCs. (**B**) Effect of rapamycin on ATP level was examined in T8993G neurons. Five-week T8993G neurons were treated with rapamycin (20 nM) and DMSO for 6 hr. The relative ATP levels of rapamycin-treated neurons were calculated as a percentage compared to the mean of DMSO-treated T8993G neurons. Bars are mean ± SD, n=3. *p<0.05 calculated by two-tailed t-test. AMPK Thr172 phosphorylation was examined by immunoblot analysis and quantified. (**C**) Five-week neurons differentiated from BJ and T8993G NPCs were used to measure protein synthesis rate. Protein synthesis are pulsed for 2 hr with $^{35}$S-Cys/Met. $^{35}$S incorporation into protein were quantified and normalized to the total protein. Data are mean ± SD, n=3. (**D**) The effect of rapamycin on glutamate-induced toxicity test. Eight-week T8993G neurons containing DCX promoter-driven GFP were treated with 100 μM glutamate in neuron growth medium with rapamycin (20 nM) or DMSO. To quantify the extent of neuronal process collapse, the number of discernible neuronal processes in a fixed photo area (350x350 pixel) were counted at time points of 0 hr and 6 hr. Bar are mean ± SD, n=3. *p<0.05 calculated by two-tailed t-test. (**E**) Immunoblot analysis of phosphorylation of ribosomal AMPK, S6, S6K and mTOR in 3-week neurons and NPCs depleting of ATP5A1. Control (Ctl) lysate were from cells infected with scramble shRNA. (**F**) Three-weeks ATP5A1-depleting BJ neurons containing DCX promoter-driven GFP. Neurons were infected with lenti-shRNA ATP5A1 and scramble shRNA from day 2 of differentiation. Relative ATP levels were quantified after normalized to protein content. Data are mean ± SD, n=3. (**G**) Three-weeks ATP5A1-depleted BJ neurons containing DCX promoter-driven GFP treated with DMSO, rapamycin and cycloheximide (200 ng/ml) for

*Figure 5 continued on next page*

*Figure 5 continued*

12 hr. (H) Quantification of beaded neuronal process. (I) Measurement of ATP level. All the experiments were repeated at least three times. (see associated *Figure 5—source data 1*).

The following source data is available for figure 5:

**Source data 1.**

glutathione and thioredoxin (*Fan et al, 2014*; *Lewis et al, 2014*; *Martínez-Reyes I and Chandel,*

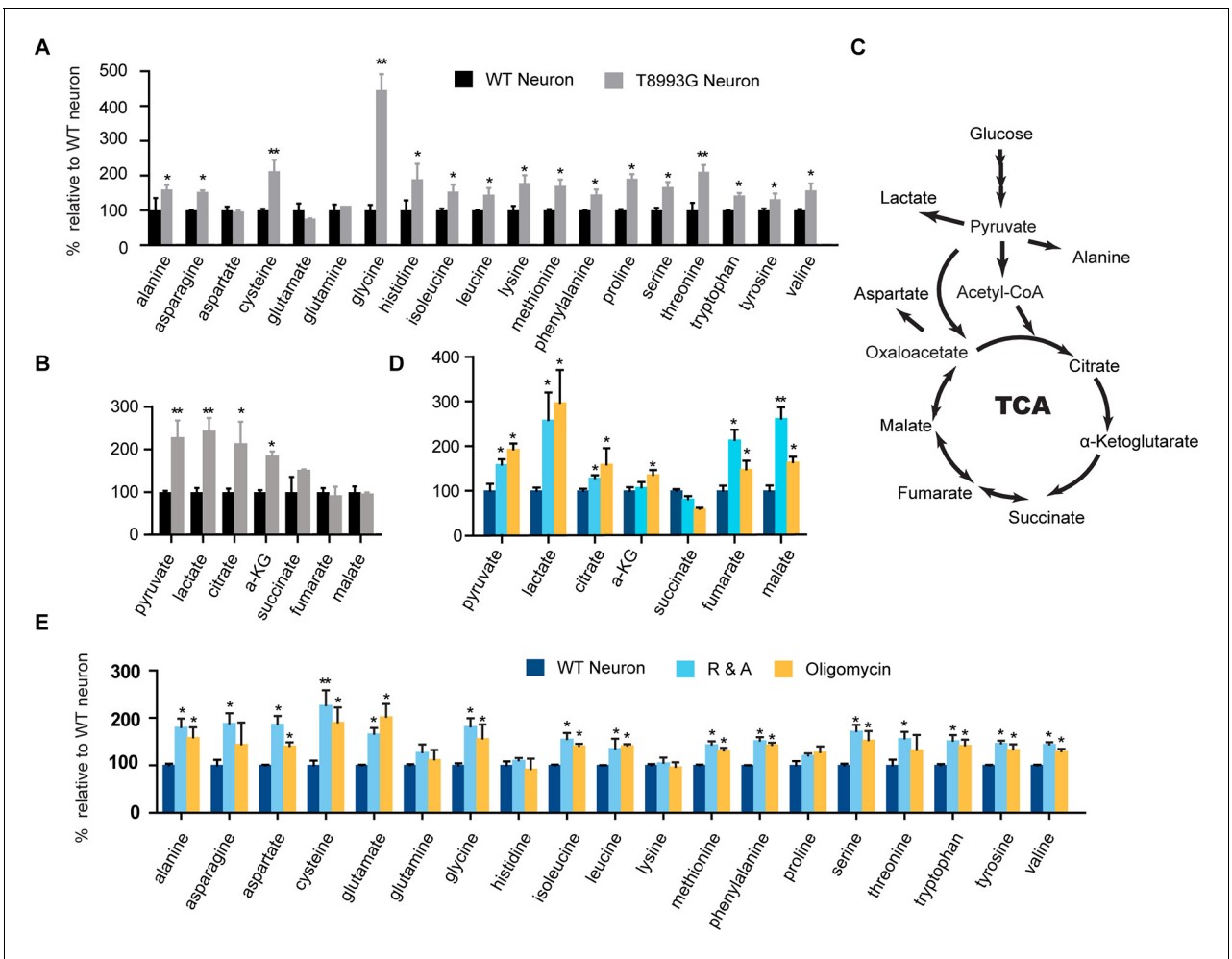

**Figure 6.** Metabolite profiling of amino acids, TCA and glycolysis intermediates. (**A**) Metabolites measured by gas chromatography mass spectrometry (GC-MS). The metabolites were extracted from 3-week T8993G and control including two BJ and one H9 neurons. Relative cellular amino acids were shown. Bar are mean ± SD, n=3. (**B**) Metabolites of glycolysis and TCA reactions. The relative amount of metabolites in T8993G neurons was presented as percentage compared to the mean of control. Bar are mean ± SD, n=3. (**C**) A simplified metabolic flow diagram of glycolysis and the TCA cycle. (**D, E**) 3-week BJ neurons were treated with oligomycin (40 nM) and rotenone and antimycin A (1 μM each) for 6 hr. Bar are mean ± SD, n=3. *p<0.05. **p<0.01, calculated by two-tailed t-test. (see associated *Figure 6—source data 1*).

The following source data and figure supplement are available for figure 6:

**Source data 1.**

**Figure supplement 1.** 10 μg protein lysate prepared form NPCs and 3-week neurons were separated on SDS-PAGE and blotted with respective antibodies.

2014), and is critical for tumor survival (*Ye et al, 2014*; *DeNicola et al, 2015*; *Kim et al, 2015*). In MILS neurons, the expression of *SHMT2, MTHFD2* and other enzymes in the *de novo* serine synthesis pathway were significantly higher than in control healthy neurons, supporting our hypothesis (unpublished data). We are currently investigating the role of this pathway in neuronal survival under mitochondrial stress. On the other hand, this accumulated glycine can be used to synthesize glutathione itself.

## Discussion

Rapamycin treatment benefits a variety of neurodegenerative diseases in animal models (*Johnson et al., 2013a*; *Lipton and Sahin, 2014*), and its effects are often attributable to the modulation of autophagy and reduction in apoptosis. A recent study reported a dramatic therapeutic effect of rapamycin on a mouse model of Leigh syndrome, deficient in NDUFS4, a component of complex I (*Johnson et al., 2013b*). The exact rescue mechanism is unclear, but increased autophagy or upregulation of mitochondrial biogenesis seem not be involved. Here, we demonstrated that rapamycin treatment significantly preserved neuronal ATP level, particularly when oxidative phosphorylation was impaired, revealing an important beneficial mechanism of rapamycin. To test its therapeutic potential on neurodegeneration resulting from energy deficiency, we developed an iPSC-based disease model of maternally inherited Leigh syndrome (MILS), an infantile neurodegenerative disease with a complex V (ATP synthase) defect due to mitochondrial DNA mutation. Rapamycin treatment significantly alleviated the ATP deficiency, reduced aberrant AMPK activation in MILS neurons, and improved their resistance to glutamate toxicity. Strikingly, ribosomal S6 and S6K phosphorylation, indicators of mTORC1 activity, were increased in neurons treated for a significant time with mitochondrial inhibitors and in MILS neurons, arguing that enhanced mTORC1 signaling might be a frequent feature of mitochondrial dysfunction in neurons.

### mTOR signaling and mitochondrial dysfunction in neurons

Our observation that in mitochondrially-defective neurons mTORC1 activity increased, rather than decreased concurrent with AMPK activation is contrary to previous reports that pharmacological disruption of mitochondrial function leads to mTORC1 inhibition due to AMPK activation by reduced energy levels (*Zoncu et al., 2011*). In proliferating NPCs, we did find that mitochondrial inhibitors activated AMPK and, consistently, decreased S6 and S6K phosphorylation. Activation of mTORC1 in neurons with mitochondrial dysfunction takes several hours, and appears to be a slow response to some cumulative effect caused by mitochondria dysfunction. This paradoxical observation could be due to unknown differences in the mTORC1 complex between NPCs and neurons. We found that AKT Ser473 phosphorylation was dramatically decreased in wild type neurons compared to NPCs, as was phosphorylation of its substrate PRAS40 at Thr246 (*Figure 6—figure supplement 1*). PRAS40 is an inhibitor of mTORC1 signaling, and its overexpression has been shown to inhibit mTORC1 hyperactivation in *Tsc2* -/- mutant cells, with AKT-mediated phosphorylation of PRAS40 preventing its inhibition of mTORC1 (*Sancak et al., 2007*; *Vander Haar et al., 2007*). An alternative explanation for mTORC1 activation emerged from our metabolite analysis, which showed that MILS neurons and neurons treated with mitochondrial drugs had significantly higher levels of amino acids, which can activate mTORC1. Indeed, it has been shown that the basal activity of S6 kinase rises progressively with increased concentration of medium amino acids in a nearly linear fashion; and at a 2-fold increased concentration, S6 kinase activity is close to maximal and no longer shows further activation by growth factors (*Hara et al, 1998*). Therefore, the increased amino acid levels in MILS neurons or mitochondria-inhibitor treated neurons are significant, and may account for the elevated mTORC1 activity. Certainly, there are other possible mechanisms; for instance, the effect might be due to decreased S6 and S6K phosphatase activity.

Similar paradoxical observations have been described in previous studies (*Ng et al., 2012*; *Nakai et al., 2015*). In an extracellular matrix detachment model, *Ng et al. (2012)* demonstrated that after detachment AMPK is activated in both K-Ras V12-transformed and non-transformed mouse embryonic fibroblasts; interestingly, mTORC1 activity only decreased, in an AMPK-dependent manner, in transformed but not non-transformed fibroblasts. They further showed that this AMPK-mediated mTORC1 inhibition decreased protein synthesis, thus preserving ATP in the detached transformed fibroblasts and delaying cell death. As in our study, rapamycin or

cyclohexmide treatment had a significant cell survival effects in their model. Notably, in the transformed fibroblasts, the level of AKT Ser473 phosphorylation was markedly stronger than in non-transformed cells, reminiscent of the situation in NPCs and neurons.

## Protein synthesis, rapamycin and neuronal ATP deficiency

When neurons were treated with rotenone and antimycin-A, protein synthesis initially dropped to ~ 36% but gradually recovered to almost 80%, correlating with the increase in S6 and S6K phosphorylation, in spite of a continuous drop in ATP levels. In the MILS neurons, S6 and S6K phosphorylation were significantly higher than in BJ and H9 control neurons. Neuronal energy deficiency caused by mitochondrial dysfunction occurs in the presence of sufficient oxygen and ample nutrition, a situation that cells in vivo rarely encounter, and there is probably no selection pressure to evolve a proper response program. We believe that the recovery of protein synthesis in such situations is an improper response that further aggravates energy deficiency. Therefore, manipulating protein synthesis to match ATP production is beneficial for neurons. Hints exist in the work of others. Overexpression of 4E-BP, a negative regulator of protein synthesis, rescues the neuronal degeneration observed in Pink1 and Parkin fly mutants with mitochondrial defects (*Tain et al., 2009*); in light of our finding, an alternative explanation is that reduction of protein synthesis is energetically beneficial for neurons with mitochondrial dysfunction. Strikingly, pathogenic LRRK2 mutation in Parkinson's disease has recently been found to induce a large increase in protein synthesis in a *Drosophila* model, and, when treated with a low-dose of the anisomycin protein synthesis inhibitor, the locomotor deficits and dopamine neuron loss in mutant LRRK2 transgenic flies were rescued (*Martin et al., 2014*). Notably, a recent genetic screen in yeast also revealed that downregulation of protein synthesis could rescue the growth of mutant cells with mitochondrial defects (*Wang and Chen, 2015*).

## ATP sparing by rapamycin is significant

As discussed in the Introduction, a cellular load of *ATP6* T8993G greater than 90% causes Leigh syndrome, a severe form of infantile neurodegeneration, whereas a 70~90% load causes a less severe neurological disease called NARP syndrome with symptoms, such as neuropathy, ataxia, and retinitis pigmentosa, gradually developing with age. In contrast, carriers with a ~50% load are generally normal, only developing late-onset cone-rod dystrophy in their forties (*Porto et al., 2001*). In a cybrid study, where patient platelets containing the T8993G mtDNA mutation were fused to human osteosarcoma cells devoid of mtDNA, mitochondrial ATP production was found to be negatively correlated with the mutation load in a nearly linear fashion (*Mattiazzi et al., 2004*). In this study, we found that rapamycin treatment of T8993G neurons increased ATP level and improved their resistance to glutamate-induced neuronal fiber collapse, a process caused by decreased intracellular ATP (*Takeuchi et al., 2005*). Therefore, a ~20% increase in ATP, the amount saved by rapamycin, which may seem small, could indeed have a significant effect on neuronal survival in Leigh or NARP syndrome patients.

## Use rapamycin to treat Leigh, NARP syndromes and other mitochondria-related neurodegenerative disorders

Leigh syndrome affects 1 in 40,000 newborns in the United States (*Darin et al., 2001*). In one fourth of the cases, these mutations occur in mitochondrial DNA (*Finsterer, 2008*; *Pinto and Moraes, 2014*). The neurological degenerative phenotypes are usually apparent in newborns, but, currently, no effective therapy is available. *Johnson et al. (2013b)* reported a dramatic therapeutic effect of rapamycin on a mouse model of Leigh syndrome. We also observed beneficial effects of rapamycin in human neurons treated with mitochondria inhibitors and in an iPSC-based disease model of maternally inherited Leigh syndrome. However, potential negative effects of rapamycin on neuronal development should not be neglected. A previous study found that focal infusion of rapamycin into dorsal hippocampus blocks axon fiber sprouting, but it should be noted that such a treatment is unable to reverse already established axon organization (*Buckmaster et al., 2009*). Therefore, we suggest that long-term usage of rapamycin for newborns should be considered with caution, but could be tried on older NARP syndrome patients to delay disease progression, and short term rapamycin usage could be considered for younger Leigh patients in emergency situations, such as fever, which often drastically and irreversibly worsens the neurodegenerative symptoms (*Uziel et al.,*

*1997*). Besides these hereditary mitochondrial diseases, mitochondrial dysfunctions are frequently observed in neurodegenerative diseases (*Lin and Beal, 2006*). Elevated p70 S6 kinase activity has been documented in the brain tissue of Alzheimer's disease patients (*An et al., 2003*). Therefore, a mild reduction in protein synthesis may be useful in other neurodegenerative diseases by increasing ATP level and simultaneously decreasing the workload of protein folding systems.

We emphasize that, whether or to what extent, our observations from a cell culture-based mitochondrial disease model reflect the in vivo situation needs further investigation using animal models with various mitochondrial deficiencies. In particular, there are some critical questions that need to be addressed under in vivo condition, e.g., the role of ATP deficiency in the neurodegenerative process; whether a reduction in protein synthesis can help balance neuronal energy expenditure and thereby delay neurodegeneration. Brain tissue is composed of mixed cell types, neurons and glial cells, which have distinct metabolic profiles and different responses to energy deficiency (*Bélanger et al, 2011*; *Almeida et al 2001*); therefore, to study the changes in ATP levels in situ and in specific cell types, an effective approach preferably at the single cell level is required. An engineered ATP fluorescent biosensor is available for cell culture systems (*Tantama et al, 2013*), which could be adapted and introduced into animal models.

## Material and methods

### Immunohistochemistry

Cells were fixed in cold 4% paraformaldehyde in PBS for 10 min. iPSCs, NPCs and neurons were permeabilized at room temperature for 15 min in 0.2% TritonX-100 in PBS. Samples were blocked in 5% BSA with 0.1% Tween 20 for 30 min at room temperature. The primary antibodies and dilutions used were: goat anti-SOX2 (Santa Cruz), 1:200; mouse anti-human Nestin (Chemicon), 1:200; rabbit anti-βIII-tubulin (Covance), 1:200; mouse anti-βIII-tubulin (Covance), 1:200; rabbit anti-cow-GFAP (Dako) 1:200; mouse anti-MAP2AB (Sigma), 1:200; secondary antibodies were Alexa donkey 488 and 568 anti-mouse, rabbit and goat (Invitrogen), used at 1:1000. Nuclear stainings were done with Hoechst (Invitrogen).

### Cell lysate preparation and immunoblotting

Cell lysates were prepared with lysis buffer containing 20 mM Tris (pH 7.5), 150 mM NaCl, 1 mM EDTA, 1 mM EGTA, 1% Triton X-100, 2.5 mM sodium pyrophosphate, 1 mM β-glycerophosphate, 1 mM $Na_3VO_4$, 1 µg/ml leupeptin. 1 mM PMSF was added immediately prior to use. The protein concentration was measured by DC protein assay (Bio-Rad). The primary antibodies and dilutions were used as follow: glycolysis antibody sampler kits (#8337&12866, Cell Signalling) used at 1:1000; the OXPHOS human WB Antibody cocktail, anti-CS and IDH2 used at 1:1000 (Abcam); anti-HSP60 and SUCLA2 (Santa Cruz); anti phospho AMPK T172 and AMPK, anti phospho ribosomal S6 Ser235/236 and S6; anti phospho mTOR Ser2481 and mTOR used at 1:1000 (Cell Signaling). Generally, 20 µg of protein lysate were loaded on SDS-PAGE gel. Immunoblotting quantification was carried out on an Odyssey Imager (Licor).

### Reprogramming iPSC

The MILS patient and BJ fibroblasts were reprogrammed into iPSCs using the standard method described by *Takahashi et al. (2007)*. GM13411 primary fibroblasts derived from a male MILS syndrome patient were obtained from the Coriell Institute for Medical Research. BJ fibroblasts were from the ATCC. Fibroblasts were cultured in DMEM media supplemented with 10% FBS, 1x Glutamax, 5 ng/ml FGF2. Fibroblasts from one well of a six-well dish were infected with retrovirus expressing *OCT4, SOX2, KLF4*, and *MYC*, and after 2 days, were split onto a 10 cm plate containing 1 million mitotically-inactivated mouse embryonic fibroblasts (mEFs). The growth medium was switched to DMEM/F12 supplemented with 20% knockout serum replacement, 1 mM L-glutamine, 0.1 mM non-essential amino acids, β-mercaptoethanol and 10 ng ml$^{-1}$ FGF2 for the 21–28 days of reprogramming. hiPSC colonies were picked and cultured onto 24-well plates coated with inactivated mEFs. hiPSCs were split through mechanic passaging with a glass pipet at early passages, while at higher passages, hiPSC could be grown on Matrigel and enzymatically digested with dispase. Karyotyping analysis was performed by Cell Line Genetics (Wisconsin, MD).

## Establishment of neuroprogenitor cells and neuron differentiation

The establishment of neural progenitor cells from iPSCs and neuronal differentiation were performed as previously described (*Brennand et al., 2011*). hESC and iPSC lines were mainly maintained on Matrigel using mTeSR1. For embryoid body formation, hESC and iPSC lines were cultured on a mitotically-inactive mouse embryonic fibroblast feeder layer in hESC medium, DMEM/F12 supplemented with 20% knockout serum replacement, 1 mM L-glutamine, 0.1 mM non-essential amino acids, β-mercaptoethanol and 10 ng ml$^{-1}$ bFGF. Neural differentiation was induced as follows: hESCs grown on inactivated mEFs were fed N2/B27 medium without retinoic acid for 2 days, and then, colonies were lifted with collagenase treatment for 1 hr at 37°C. The cell clumps were then transferred to ultra-low attachment plates. After growth in suspension for 1 week in N2/B27 medium, aggregates form embryoid bodies, which were then transferred onto polyornithine (PORN)/laminin-coated plates and developed into neural rosettes in N2/B27 medium. After another week, colonies showing mature neural rosettes with biopolar neuroprogenitor cells migrating out from the colony border, were picked under a dissecting microscope, digested with accutase for 10 min at 37°C and then cultured on polyornithine(PORN)/laminin-coated plates in N2/B27 medium supplemented with FGF2.

For neuron differentiations, neuroprogenitor cells were dissociated with accutase and plated in neural differentiation media, 500 ml DMEM/F12 GlutaMAXTM, 1x N2, 1X B27+RA, 20 ng/ml BDNF (Peprotech), 20 ng/ml GDNF (Peprotech), 200 nM ascorbic acid (Sigma), 1 mM dibutyrl-cyclicAMP (Sigma) onto PORN/Laminin-coated plates. For one well of a 6-well plate, 200,000 cells/well were seeded; for one well of a 12-well plate, 80,000 cells were seeded. Neurons can be maintained for 3 months in a 5% $CO_2$ 37°C incubator.

## qRT-PCR

Total RNA was isolated using RNeasy kit (QIAGEN). 500 ng of total RNA from each sample was used for cDNA synthesis by MMLV reverse transcriptase; and quantitative real-time polymerase chain reaction (PCR) was performed with SYBR Green Master Mix on ABI 7000 cycler (Applied Biosystems) and normalized to β-actin. Primer sequences were referred from qPCR primerDepot (http://primerdepot.nci.nih.gov/).

## FACS measurement of mitochondrial membrane potential and cellular ROS

Mitochondrial membrane potential was measured by flow cytometry of iPSCs, NPCs and neurons stained with TMRE (Invitrogen). Cells were dissociated with accutase, spun down at 350 g for 10 min, and then, resuspended in PBS with 2% bovine serum albumin (BSA) loaded with 10 nM TMRE for 15 min at 37°C. The cells were washed again, filtered through a 250-μM nylon sieve and kept in PBS on ice. The TMRE signal was quantified using the FL2 channel of a Becton Dickinson FACScan. Each set of measurements included a control sample pretreated for 30 min with 20 μM of CCCP, a mitochondrial uncoupler, to abolish mitochondria membrane potential. Data were analyzed using FloJo; and the mean value was used to compare the mitochondria potential between BJ and T8993G cells. Similarly, cellular ROS level was measured by 10 μM CM-H2DCFDA (Invitrogen) staining for 30 min and detected in the FL1 channel.

## OCR, ATP and Lactate measurement

The OCR of iPSC, NPCs and neurons grown in Seahorse plates was measured using an extracellular Flux Analyzer (Seahorse Bioscience), following the manufacturer's instructions. After the measurement, cells were lysed in 60–100 μl lysis buffer with two "freeze and thaw" cycles on dry ice. Protein concentrations were determined by DC protein assay (Bio-Rad). The OCR values were normalized by protein mass. For measurement of cellular ATP content, neurons were lysed directly on plates with protein extract buffer by two freeze-and-thaw cycles in dry ice. The ATP content was quantified by CellTiter-Glo Luminescent Cell Viability/ATP Assay kit (Promega), and normalized by protein content measured by DC protein assay (Bio-Rad). For measurement of secreted lactate levels, medium from iPSCs, NPCs and neurons was freshly changed and collected after 12 hr, and cells were frozen on the plate and lysed by two freeze-and-thaw cycles on dry ice. Medium lactate was measured using the Lactate Assay kit (BioVision) and normalized by total protein content.

## Protein synthesis measurement

To avoid the medium change effect on mTOR signaling, 10 μl (100 μCi) Express $^{35}$S protein labelling mix (Perkin Elmer) were added into NPCs or neurons grown in 12 well plate containing 1 ml NPC or neuronal growth medium. Cells were labelled for 2 hr and lysed on plate after two times of PBS wash. Twenty-five μl lysate were mixed with 5 μl 100% TCA, incubated on ice for 30 min, and spotted on Whatman 3MM filter paper. The filter papers were washed twice with cold 5% TCA and air-dried. The radioactivity was determined by scintillation counting.

## Metabolite analysis

Neurons were grown in a 6-well plate. After growth in fresh medium for 12 hr, cells were washed quickly 3 times with cold PBS, and 0.45 ml cold methanol (50% v/v in water with 20 μM L-norvaline as internal standard) was added to each well. Culture plates were transferred to dry ice for 30 min. After thawing on ice, the methanol extract was transferred to a microcentrifuge tube. Chloroform (0.225 ml) was added, the tubes were vortexed and centrifuged at 10,000 g for 5 min at 4°C. The upper layer was dried in a centrifugal evaporator and derivatized with 30 μl O-isobutylhydroxylamine hydrochloride (20 mg/ml in pyridine, TCI) for 20 min at 80°C, followed by 30 μl N-tert-butyldimethyl-silyl-N-methyltrifluoroacetamide (Sigma) for 60 min at 80°C. After cooling, the derivatization mixture was transferred to an autosampler vial for analysis. GC-MS analysis was performed in the Cancer Metabolism core at the Sanford-Burnham Medical Research Institute (La Jolla, California). More details including the parameters of machine settings can be found in the publication from the center (*Scott et al., 2011*).

## Statistical analysis

Comparisons were done by Student's t-test. Statistical analyses were performed using GraphPad Prism.

## Acknowledgements

We thank members of the Hunter lab for helpful discussions, and Jill Meisenhelder, Suzy Simon and Justin Zimmerman for laboratory support. This study was supported by NIH grants (CA14195, CA80100 and CA82683) to TH, and by the G Harold & Leila Y Mathers Charitable Foundation, the JPB Foundation, the Leona M and Harry B Helmsley Charitable Trust grant #2012-PG-MED002, Annette Merle-Smith, CIRM (TR2-01778) to FHG, and by NIH grant (DK057978) to RME. The study was supported by Salk core facilities and their staff including the Stem Cell, Advanced Biophotonics, Flow Cytometry and Functional Genomics Cores, and by the Helmsley Center for Genomic Medicine. FHG is the Vi and John Adler Chair for Research on Age-Related Neurodegenerative Disease. TH is a Frank and Else Schilling American Cancer Society Professor, and the Renato Dulbecco Chair in Cancer Biology. XZ was supported by a fellowship from the California Institute of Regenerative Medicine and a Salk Pioneer postdoctoral fellowship.

## Additional information

### Competing interests

TH: Senior editor, *eLife.* The other authors declare that no competing interests exist.

### Funding

| Funder | Grant reference number | Author |
|---|---|---|
| National Institutes of Health | DK057978 | Ronald M Evans |
| California Institute for Regenerative Medicine | TR2-01778 | Fred H Gage |
| National Institutes of Health | CA14195 | Tony Hunter |
| National Institutes of Health | CA80100 | Tony Hunter |
| National Institutes of Health | CA82683 | Tony Hunter |

The funders had no role in study design, data collection and interpretation, or the decision to submit the work for publication.

## Author contributions

XZ, Planned and supervised the project, Established MILS and BJ iPSCs, Established neural progenitor cells, neuron differentiation and performed related assays, Analyzed metabolic phenotypes and rapamycin's effects on neurons, Acquisition of data, Analysis and interpretation of data, Drafting or revising the article; LB, Planned and supervised the project, Established neural progenitor cells, neuron differentiation and performed related assays, Acquisition of data, Analysis and interpretation of data, Drafting or revising the article; MJ, Established neural progenitor cells, neuron differentiation and performed related assays, Performed iPSCs and neuronal cell culture and FACS analysis, Acquisition of data, Analysis and interpretation of data, Drafting or revising the article; YK, Supervised mitochondria function assays, Analysis and interpretation of data, Contributed unpublished essential data or reagents; WF, Supervised mitochondria function assays, Contributed their knowledge of metabolism and mitochondrial DNA diseases, Acquisition of data, Analysis and interpretation of data, Drafting or revising the article; CB, Performed electrophysiological analysis, Acquisition of data, Analysis and interpretation of data, Drafting or revising the article; TB, Supervised iPSC reprogramming, Analysis and interpretation of data, Drafting or revising the article, Contributed unpublished essential data or reagents; RME, Contributed their knowledge of metabolism and mitochondrial DNA diseases, Conception and design, Drafting or revising the article; FHG, Planned and supervised the project, Drafting or revising the article; TH, Conception and design, Drafting or revising the article

## Author ORCIDs

Cedric Bardy, http://orcid.org/0000-0001-8321-0852
Tony Hunter, http://orcid.org/0000-0002-7691-6993

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
