## [Decision Letter]

Thank you for submitting your work entitled "Alleviation of neuronal ATP deficiency by mTOR inhibition as a treatment for mitochondria-related neurodegeneration" for consideration by *eLife*. Your article has been reviewed by two peer reviewers, and the evaluation has been overseen by a Reviewing Editor and Harry Dietz as the Senior Editor.

The reviewers have discussed the reviews with one another and the Reviewing Editor has drafted this decision to help you prepare a revised submission.

This paper provides compelling evidence that inhibition of mTOR activity with rapamycin maintains ATP levels and thereby ameliorates neurological degeneration in cell culture models of mitochondrial disease. Moreover, they attribute some of the beneficial effect of rapamycin treatment to reduction is protein synthesis mediated by reduction in S6 phosphorylation. Consequently, they propose that rapamycin and/or mild inhibition of protein synthesis may be therapeutically useful for treating mitochondrial diseases. These cell culture results are encouraging; however, it is imperative that the treatment strategies be tested in animal models of various mitochondrial deficiencies before proposing therapeutic approaches.

Essential revisions:

Since it is unrealistic for the authors to add therapeutic testing to this paper, they should acknowledge the need to test their ideas in existing animals’ models and/or develop appropriate models to test whether inhibition of mTOR or protein synthesis are universally beneficial and assess the mechanisms involved in vivo.

*Reviewer #1:*

This study by Zheng et al. builds on prior work indicating that mTOR is hyperactivated in response to certain forms of mitochondrial dysfunction and inhibition of mTOR can improve outcome in animal and cell based models of mitochondrial disease. Specifically, the authors show that cells treated with mitochondrial poisons show increased mTOR signaling and reduced ATP levels that can be rescued by rapamycin treatment. They further show that inhibition of protein synthesis in these cells can partially phenocopy rapamycin by rescuing ATP levels. They go on to describe an iPSC model of Leigh Syndrome in which they observe increased mTOR signaling. They show that treatment with rapamycin can improve some cellular phenotypes, in particular rescuing ATP levels, and propose that this is mediated through inhibition of protein synthesis.

Overall, this study provides important additional evidence for the idea that mTOR inhibitors may be of therapeutic value for treatment of mitochondrial disease. It also supports a specific model that rescue of mitochondrial defects by rapamycin is mediated in part by reducing protein synthesis, a particularly ATP consuming process, and thereby rescuing ATP deficits in cultured cells.

My main question about this study is the extent to which these observations are relevant in a whole animal or person. It is not clear that ATP deficiency is a major driver of disease in mitochondrial disease, nor is it clear that rapamycin treatment significantly reduces protein synthesis in vivo. I can imagine that the high nutrient growth conditions used for cell culture create a somewhat artificial system in which protein synthesis and ATP consumption by protein synthesis are elevated relative to any physiologically relevant situation. In this regard, it may be worth noting that the *NDUFS4* study by Johnson et al. did not detect a significant difference in ATP levels in KO compared to WT animals nor was a change in ATP levels in whole brain upon rapamycin treatment observed. Obviously, this negative result from Johnson et al. is not conclusive, but it would greatly enhance the relevance of this study if the authors could offer some direct experimental evidence that their proposed mechanism is important in a mouse model. At a minimum, the possibility that the importance of the translation/ATP mechanism proposed here is enhanced in cell culture relative to tissues in vivo should be addressed in the text.

*Reviewer #2:*

This is a very interesting manuscript uncovering an unanticipated pathway of mTOR activation in neurons as a consequence of mitochondrial stress induced chemically or genetically. This mTOR activation in context of mitochondrial insufficiency leads to ATP depletion and neurotoxicity and may explain some of the neurological features of mitochondrial diseases. This pathway in differentiated neurons is absent in neural progenitor cells. Inhibition of mTOR with rapamycin or inhibition of protein synthesis with cycloheximide rescue the ATP depletion and cell death suggesting some new rationales for therapeutic intervention. This is a thorough and well-controlled study appropriate for *eLife*.

Comments:

In the subsection “Shutoff of aerobic glycolysis during neuronal differentiation exposes ATP synthesis deficiency in T8993G MILS neurons”: "Without LDHA and HK2, neurons cannot[…]" – cannot seems too strong. If the authors reintroduce LDHA and/or HK2 and this rescues ATP levels the statement would be justifiable. As it stands "cannot" would be better stated something like "appear unable". The reintroduction would be an interesting experiment to perform.

In the first paragraph of the subsection “Rapamycin treatment significantly increases ATP level and decreases aberrant AMPK phosphorylation in T8993G MILS neurons”: T8993G mutant neuron lines all nicely show increased S6 phosphorylation (Figure 5). Does rapamycin treatment inhibit this?

---

## [Author Response]

Essential revisions:

*Since it is unrealistic for the authors to add therapeutic testing to this paper, they should acknowledge the need to test their ideas in existing animals’ models and/or develop appropriate models to test whether inhibition of mTOR or protein synthesis are universally beneficial and assess the mechanisms involved in vivo. Reviewer #1: My main question about this study is the extent to which these observations are relevant in a whole animal or person. It is not clear that ATP deficiency is a major driver of disease in mitochondrial disease, nor is it clear that rapamycin treatment significantly reduces protein synthesis in vivo. I can imagine that the high nutrient growth conditions used for cell culture create a somewhat artificial system in which protein synthesis and ATP consumption by protein synthesis are elevated relative to any physiologically relevant situation. In this regard, it may be worth noting that the Ndufs4 study by Johnson et al. did not detect a significant difference in ATP levels in KO compared to WT animals nor was a change in ATP levels in whole brain upon rapamycin treatment observed. Obviously, this negative result from Johnson et al. is not conclusive, but it would greatly enhance the relevance of this study if the authors could offer some direct experimental evidence that their proposed mechanism is important in a mouse model. At a minimum, the possibility that the importance of the translation/ATP mechanism proposed here is enhanced in cell culture relative to tissues in vivo should be addressed in the text.* We appreciate and agree with the reviewer's point that the cell culture system may not fully reflect the in vivo situation. Going forward it will indeed be important to examine whether rapamycin treatment or reduction of protein synthesis is able to preserve neuronal ATP levels and show beneficial effects in animal models of mitochondrial diseases, but it would be hard to generate a model of MILS for mitochondrial DNA mutant, and we feel that this is beyond the scope of the present study. To acknowledge the limitations of the current study and highlight the issues raised by the reviewer, we have added the following paragraph in the Discussion.

"We emphasize that, whether or to what extent, our observations from a cell culture-based mitochondrial disease model reflect the in vivo situation needs further investigation using animal models with various mitochondrial deficiencies. […] An engineered ATP fluorescent biosensor is available for cell culture systems (Tantama et al., 2013), which could be adapted and introduced into animal models. "

*Reviewer #2: Comments: In the subsection “Shutoff of aerobic glycolysis during neuronal differentiation exposes ATP synthesis deficiency in T8993G MILS neurons”: "Without LDHA and HK2, neurons cannot[…]" – cannot seems too strong. If the authors reintroduce LDHA and/or HK2 and this rescues ATP levels the statement would be justifiable. As it stands "cannot" would be better stated something like "appear unable". The reintroduction would be an interesting experiment to perform.*

We would like to thank the reviewer for highlighting the importance of our work.

We were also intrigued by the disappearance of HK2 and LDHA during neuronal differentiation and the functional significance of their loss. In a separate project to investigate metabolic changes during differentiation of neural progenitor cells into neurons we analyzed the effect of constitutive expression of HK2 and LDHA during neuronal differentiation, and these data will be included in another manuscript describing the metabolic transition from neural progenitor cells to post-mitotic neurons. The data are shown in Author response image 1. In brief, we stably transduced neural progenitor cells (NPCs) with constitutively expressed HK2 and LDHA; ~80% of NPCs had discernible expression and proliferation appeared normal. However, constitutive HK2 and LDHA expression during neuronal differentiation led to extensive cell death and significantly increased GFAP-positive glial cells from 4% of total cells in the control NPC cells containing expression vector alone to ~40% of total cells. The LDHA signals were readily detected in GFAP-positive glial cells but not in MAP2-positive neurons. Importantly, the dead cells signified by condensed nuclei were associated with punctate staining of LDHA and MAP2, a neuronal marker, indicating that neurons were sensitive to HK2 and LDHA re-expression. At the moment, we do not fully understand the mechanism behind this phenomenon. We agree with the reviewer's point, and have changed the phrase according to their suggestion.

The effects of HK2 and LDHA re-expression on neuronal differentiation. (**A**) Immunoblotting analysis of HK2 and LDHA in NPCs and 3-week neurons constitutively expressing HK2 and LDHA (Flag-tagged). (**B**) Immunostaining of NPCs with anti-FLAG antibody (green), and nuclear staining was done with Hoechst (red). The percentage of Flag-positive were quantified, and 100 cells were counted for each group. (**C**) Immunostaining analysis of MAP2 and GFAP in 3-week neurons. The percentage of GFAP and MAP2 cells were quantified, and 100 cells were counted for each group, and three times of neuronal differentiation were included. Bars are mean ± SD, n=3. The GFAP mRNA abundance in the RNA extracted from neuronal culture was quantified by real time PCR and normalized to β-actin, and presented by fold increase compared to neurons differentiated from control NPCs. Bars are mean ± SD, n=3. (**D**) Nuclear staining with Hoechst in NPCs and 3-week neurons. The percentage of condensed nuclear were quantified, and 50 cells were counted for each group. Bars are mean ± SD, n=3. (**E**) Immunostaining analysis of LDHA, MAP2 and GFAP in 3-week neurons differentiated from NPC constitutively expressing HK2 and LDHA. (F, G) Co-localization of irregular puntated staining of LDHA (green) with MAP2 (red, in F) or condensed nuclear staining with Hoechst (red, in G) in 3-week neurons differentiated from NPC constitutively expressing HK2 and LDHA.

In the first paragraph of the subsection “Rapamycin treatment significantly increases ATP level and decreases aberrant AMPK phosphorylation in T8993G MILS neurons”: T8993G mutant neuron lines all nicely show increased S6 phosphorylation (Figure 5). Does rapamycin treatment inhibit this?

Rapamycin treatment effectively decreased S6 phosphorylation in MILS neurons. The new data were added into Figure 5.